# MOFs-Derived Nano-CuO Modified Electrode as a Sensor for Determination of Hydrazine Hydrate in Aqueous Medium

**DOI:** 10.3390/s20010140

**Published:** 2019-12-24

**Authors:** Yaqi Lu, Dan Wu, Ziyin Li, Quanjie Lin, Xiuling Ma, Zhangjing Zhang, Shengchang Xiang

**Affiliations:** 1Fujian Provincial Key Laboratory of Polymer Materials, College of Chemistry and Materials Science, Fujian Normal University, 32 Shangsan Road, Fuzhou 350007, China; fjluyaqi@163.com (Y.L.); lyxylyh@163.com (D.W.); lzyinann@163.com (Z.L.); linquanjie@hotmail.com (Q.L.); zzhang@fjnu.edu.cn (Z.Z.); 2College of Chemistry and Materials Science, Longyan University, No.1 North Dongxiao Rd., Longyan 364012, China

**Keywords:** metal-organic frameworks, pyrolysis derivatives, electrochemical sensor, hydrazine hydrate

## Abstract

It very important to be able to efficiently detect hydrazine hydrate in an aqueous medium due to its high toxicity. Here, we have proposed a new idea: to construct a sensor for the rapid determination of hydrazine hydrate based on the nano-CuO derived by controlled pyrolysis of HKUST-1 [Cu_3_(BTC)_2_(H_2_O)_3_]. The as-prepared CuO at 400 °C possesses a uniform appearance with nano-structure via SEM images, and the nano-CuO-400 has exhibited excellent electrocatalytic activity towards hydrazine oxidation. Amperometric *i-t* curves shows the peak current as linearly proportional to the hydrazine concentration within 1.98–169.3 μmol L^−1^ and 232–2096 μmol L^−1^ with the detection limit of 2.55 × 10^−8^ mol L^−1^ and 7.01 × 10^−8^ mol L^−1^, respectively. Moreover, the sensor constructed in the experiment shows good selectivities, and it is feasible to determining actual water samples.

## 1. Introduction

Hydrazine hydrate, which is an important chemical raw material, is widely used as antioxidant, rocket propellant, preservative, insecticide, plant growth regulator, etc. [1]. However, it is highly toxic and a hydrazine content above 10 ppb can induce steatosis, nausea, pulmonary edema, nasal irritation, DNA damage, temporary blindness and so on [2,3]. In addition, hydrazine hydrate has been listed as a carcinogen by the US Environmental Protection Agency [4]. Therefore, it is of great significance to detect hydrazine hydrate efficiently and sensitively.

Various methods for detecting hydrazine hydrate have already been established, such as spectrophotometry, electrochemiluminescence, GC-MS, and high-performance liquid chromatography (HPLC) [5,6,7,8,9], which display excellent repeatability and sensitivity. However, they may be inconvenient due to the high-cost and time-consuming running. Therefore, it is worth developing a simple and practical approach for determining the trace levels of hydrazine in water. Electrochemical techniques offer portable, quick, and economical methodologies, which has attracted the attention of researchers [10]. As hydrazine has strong reducibility, it is easy to oxidize on the surface of electrodes, and the electrochemical determination of hydrazine hydrate has been proven to be a feasible detection method [11]. To improve the sensitivity and reproducibility of hydrazine’s reaction on electrodes, some functionalized nanomaterials have been used for the construction of hydrazine hydrate chemical sensors [12,13,14,15,16,17]. Although the above method enhances the electrochemical reaction of hydrazine, most are noble metal nanoparticles, which limits the application in reality. Cupric oxide, a non-precious metal oxide, has been widely used in sensors due to its low cost and good catalytic ability [18,19,20], and it has also been reported as useful for detecting hydrazine hydrate [21,22,23,24,25]. Further study on the sensor is still required in order to accelerate development of an effective method for detecting hydrazine hydrate.

Metal-organic frameworks (MOFs), which have attracted extensive attention in recent years, are a kind of inorganic-organic hybrid materials with extensive applications including gas adsorption and separation, catalysis, luminescence, electrode materials, etc. [26,27,28,29,30]. Porous MOF-based sensors were also developed [11]. However, most MOFs are unstable in a water phase, so it is hard to apply them directly to electrochemical measurement in aqueous environment. Due to the homogeneity of MOFs, the metal and organic ligands in MOFs can be transformed into derivatives with uniform structure and stable chemical properties under certain conditions, which attracted considerable attentions, especially in electrochemistry application [31,32,33].

In this study, instead of noble metal nanoparticles, the nano-CuO derived from the pyrolysis of HKUST-1, which is one of the first reported MOFs, was used to construct the hydrazine hydrate sensor (shown in Figure 1). HKUST-1[Cu_3_(BTC)_2_(H_2_O)_3_] was chosen as the template for the pyrolysis derivatives because its skeleton contains coordination unsaturated metal Cu (II) and it easily obtains nano-CuO. Additionally, it has been discovered that the structures and electrocatalytic performances of the materials may be controlled by the pyrolysis temperature. The response of hydrazine hydrate in the nano-CuO-400 sensor is superior to that of the CuO-300 sensor, CuO-500 sensor, or bulk-CuO sensor. Furthermore, the nano-CuO-400-coated electrode with a low detection limit of 2.55 × 10^−8^ mol L^−1^ is approximately 63 times lower than that of the CuO-hollow-spheres-coated one [24] and equal to the nano-copper oxide obtained by electrodeposition [26], in which is difficult to obtain a lot of stable nano-sized cupric oxide. These results might confirm that the nano-CuO derived from the pyrolysis of MOFs could effectively improve the detection limit of the hydrazine hydrate sensor in aqueous phase. In addition, based on CuO nanostructures possessing superior physical and chemical properties, they show promising applications in various fields, such as gas sensors, nanohybrid catalyst and so on [34,35,36]. We believe the as-prepared nano-CuO-400 should has potential applications in other electrocatalytic fields.

## 2. Materials and Methods

### 2.1. Instruments and Reagents

Powder X-ray diffraction (PXRD) was performed over the 2θ range of 5°–30° or 5°–80° (MiniFlex II Xpert power diffractometer equipped with a Cu-sealed tube at 40 kV and 40 mA, Rigaku, Woodlands, TX, USA). Fourier transform infrared spectra (FTIR, KBr pellets) were recorded in the range of 400~4000 cm^−1^ on a 5700 FT-IR spectrometer (Thermo Nicolet, Austin, USA). Thermal gravimetric analysis (TGA) curves were obtained from 30 to 600 °C at the heating rate of 10 °C/min under nitrogen atmosphere by using a TGA/STDA 851 thermal analyzer (Mettler, Zurich, Switzerland). The morphology of the products was characterized with a S-4800 scanning electron microscope (SEM, Hitachi, Tokyo, Japan). Electrochemical measurements were conducted on an electrochemical workstation (CHI660E, Shanghai Chenhua Apparatus Company, Shanghai, China). Copper nitrate hydrate (Cu(NO_3_)2·3H_2_O), *N*,*N*-dimethyl-formamide (DMF), acetone, sodium hydroxide (NaOH, 96%) and hydrazine hydratex(N_2_H_4_·H_2_O) were purchased from Sinopharm Chemical Reagent Co., Ltd. (Shanghai, China). 1,3,5-Benzenetricarboxylic acid (99%, H3BTC) was purchased from Aladdin (Shanghai, China). All the other chemicals were analytical reagent grade or better.

### 2.2. Preparation of Nano-CuO

The quartz boat was covered with the prepared HKUST-1 (see the Appendix A). The heat treatment was carried out under the air condition in a tubular furnace. HKUST-1 were heated to 400 °C at a rate of 5 °C/min, and at 400 °C for 2 h, then naturally cooled to room temperature. The product is named CuO-400. In order to study the effect of heat treatment temperature on the product, with other conditions unchanged, the products named CuO-300 and CuO-500 were obtained by heat treatment at 300 °C and 500 °C, respectively.

### 2.3. Fabrication of Nano-CuO Sensor

A glassy carbon electrode (GCE) was polished carefully with 0.3 cm and 0.05 cm alumina slurry, and thoroughly rinsed ultrasonically with acetone and water for 2 min in turn, then dried under infrared light. Took 5 mg CuO-400 into 1 mL ethanol. After improving the uniformity by the ultrasonic dispersion, added 10 μL 5 mg/mL CuO-400 suspension on the GCE surface using drop-casting method. Then dried the electrode by infrared light. Finally, 5 μL of 0.5% Nafion droplets were applied to the surface of the electrode, and CuO-400/GCE was obtained. The preparation of CuO-300/GCE or CuO-500/GCE was done following the above steps.

### 2.4. Electrochemical Detection

The detection was conducted by a three-electrode system in which a modified GCE (3.0 mm in diameter), Ag/AgCl electrode with saturated KCl and a platinum electrode were used as working electrode, reference electrode and the counter electrode, respectively. Cyclic voltammetry (CV) was performed in the static solution, and the potential range was −0.4 V to 0.8 V. The amperometric *i*-*t* curve was obtained through dynamic tests that required constant stirring of the solution. The supporting electrolyte used was 0.1 mol L^−1^ NaOH. Before each test, the solution was flushed with nitrogen for 15 min to ensure that there was no oxygen interference in the solution.

Water samples collected from the Minjang river (in Fujian, China) were filtered three times before analysis. Different amounts of hydrazine hydrate were added to the water samples which were then analyzed under optimized conditions using the standard addition method.

## 3. Results and Discussion

### 3.1. Characterization of HKUST-1

PXRD was used to evaluate whether the synthesized compound (HKUST-1) was a pure phase. Appendix A showed the pattern of the synthesized HKUST-1, which was consistent with the simulated PXRD pattern of HKUST-1 reported previously. The results showed that the synthesized HKUST-1 is pure. The FT-IR spectra of HKUST-1 was shown in Appendix A. The characteristic bands at 1643 cm^−1^ and 1372 cm^−1^ are attributed to the asymmetric stretching vibration and symmetric stretching vibration of the carboxyl groups on trimesic acid, respectively. The broad band at about 3430 cm^−1^ could be assigned to O−H stretching vibrations. Meanwhile, the thermal behavior of HKUST-1 powders was investigated by TGA technique which can provide the base for the pyrolysis characteristics of HKUST-1, shown in Appendix A. Loss of weight before 100 °C was caused by water adsorbed on the pore of samples, while the loss of the weight at 210 °C–337 °C was related to the removal of the DMF solvent and after 337 °C, it was ascribed to the decomposition of HKUST-1. Thereby, 300 °C, 400 °C and 500 °C were chosen as the pyrolysis temperatures for obtaining nano-CuO.

### 3.2. Analysis of Nano-CuO

The PXRD patterns of the nano-CuO derived by the pyrolysis of HKUST-1 at the different heat treatment temperatures are shown in Figure 2A. The diffraction peaks of CuO-400 or CuO-500 at 32.5°, 35.5°, 38.7°, 48.8°, 53.4°, 58.3°, 61.6°, 66.3°, 68.0°, 72.3° and 74.9° matched perfectly with the peaks of the card of JCPDS No. 41–0254, which were found to be in agreement with the (110), (002), (111/200), (−202), (020), (202), (−113), (−311), (220), (311) and (−222) facet of CuO crystal [37], respectively. Meanwhile, CuO-300 didn’t show all the feature peaks because of incomplete decomposition of HKUST-1 at 300 °C. What’s more, the influence of the different pyrolysis temperature on the electrochemical property of hydrazine hydrate on nano-CuO/GCE were further examined by the CV in 0.1 mol L^−1^ NaOH solutions with 1 mmol L^−1^ hydrazine hydrate, shown in Figure 2B. The highest peak current as well as good shape of hydrazine hydrate were displayed on the electrode modified with CuO-400. Although the CuO-400 and CuO-500 presented the same diffraction pattern, their electrochemical performance were different. These indicated that different heat treatment temperature could affect the properties of materials. Therefore, CuO-400 was selected as the main subject in the following section.

SEM studies were further conducted to investigate microstructure morphology of HKUST-I and its pyrolysis CuO-400, as shown in Figure 3. It can be seen that the most of HKUST-I is octahedral morphology and has smooth surface as well as good dispersion, while CuO-400 lost the original appearance of HKUST-I and became irregular. Meanwhile, its particle size is less than 500 nm.

### 3.3. Optimization of the Experimental Conditions

To get the optimal conditions for detecting hydrazine, CV was performed in the following sections by using nano-CuO-400/GCE. The effect of the drop-coating doses of nano-CuO-400 was performed with different doses of nano-CuO-400 coating on GCE, shown in Figure 4A. From 6 μL to 10 μL, the responses to 1 mmol·L^−1^ hydrazine hydrate increased dramatically. After the dose was increased to 12 μL, the responses gradually decreased. Therefore, the drop amount of nano-CuO-400 was selected to be 10 μL in this study. Also, the effect of the concentration of NaOH solution was studied, as shown in Figure 4B. The peak current of hydrazine increased when increasing NaOH concentration (0.05 mol·L^−1^–0.1 mol·L^−1^), and the oxidation peak potential showed negative shift. After increasing the concentration of NaOH (over 0.1 mol·L^−1^), the oxidation peak current decreased, and the oxidation peak potential shifted positively. A more negative oxidation peak potential and higher peak current may be beneficial for faster electron-transfer reaction and sensitive detection, respectively. Thus, 0.1 mol·L^−1^ NaOH was considered the appropriate concentration for investigating the electrochemical activity of hydrazine. Additionally, the oxidative peak potential of hydrazine shifted with the changes of the NaOH concentration, which indicated that the rate-determining step is the formation of the [HO–H_2_NNH_2_–OH]^2−^ ion [26].

### 3.4. Electrochemical Properties of Nano-CuO-400/GCE Sensor

To analyze the charge transfer resistance of the nano-CuO/GCE, the electrochemical impedance spectroscopy (EIS) was conducted in this work, shown in Figure 5A. Nano-CuO-400/GCE showed a lower resistance (~100 Ω) than bare GCE (~370 Ω) in 5.0 mmol·L^−1^ [Fe(CN)_6_]^3/4−^ with 0.1 mol·L^−1^ KCl, which suggested that electron transfer at the nano-CuO-400/GCE is faster than that at the bare GCE. Figure 5B shows the effect of scan rates (v) on the peak current (I) in 0.1 mol·L^−1^ NaOH containing 1 mmol·L^−1^ hydrazine. The peak current increased at the scan rate of 20 mV/s~400 mV/s, and a linear relationship that Ip = 0.043 *v*^1/2^ + 0.1104 (R^2^ = 0.998, inset) between I and *v*^1/2^ was obtained. This result indicates that the oxidation process of hydrazine hydrate at the sensor is a diffusion-controlling behavior.

In addition, HKUST-1 and bulk-CuO were selected as references for controlled trials. The results of CV for different materials at the scan rate of 50 mV/s are shown in Figure 6A–D.

There are no detectable peak in 0.1 mol·L^−1^ NaOH solution (curve a) at all electrodes. When the solution containing 1 mol·L^−1^ hydrazine hydrate (curve b), different electrodes showed different results. The oxidation peak of hydrazine hydrate was not observed on the bare GCE in Figure 6A (curve b), indicating that the GCE has no catalytic activity to hydrazine hydrate. To the HKUST-1/GCE in Figure 6B (curve b), there was a wide, weak and irreversible oxidation peak at ~0.6 V, while, a sharp and strong oxidation peak appears at ~0.31 V on the CuO-400/GCE in Figure 6C (curve b). Additionally, the hydrazine hydrate showed an irreversible oxidation peak at ~0.5 V for the purchase of nano-CuO (Figure 6D, curve b).

Comparing CuO-400/GCE, bulk-CuO/GCE and HKUST-1/GCE in Figure 6E, the catalytic oxidation at the CuO-400/GCE are much sharper and shift to a negative-going direction, which reflects a faster electron-transfer reaction on the CuO-400/GCE owing to the high catalytic effect of the characteristic of nano-CuO. In addition, CV was performed to determine hydrazine hydrate after the CuO-400/GCE sensor was immersed in solutions containing hydrazine hydrate of different concentration (0, 150, 200, 400, 800 μmol·L^−1^), shown in Figure 6F. It can be found that the oxidation peak current increases with the increase of hydrazine hydrate concentration in the range of 0~800 μmol·L^−1^. This proved that the nano-CuO-400 shows a good electrocatalyst to hydrazine hydrate.

### 3.5. The Detection Limit of the Sensor for Hydrazine Hydrate

In order to further evaluate the electrochemical sensing for hydrazine hydrate on the CuO-400/GCE sensor, amperometric *i*-*t* curves were performed at an applied potential of 0.31 V with the successive addition of hydrazine ranging from 1.98 μmol·L^−1^ to 2096 μmol·L^−1^ in 0.1 mol·L^−1^ NaOH by stirring. As shown in Figure 7a, the peak current gradually increased with the increase of the hydrazine hydrate concentrations. It appeared to platform at every concentration stage and reached a stable current for no more than 5 s, indicating that the sensor has a short response time. Figure 7b shows the peak current as linearly proportional to the hydrazine concentration in the range of 1.98 μmol·L^−1^ to 169.3 μmol·L^−1^ with the linear regression equation: I (μA) = 0.08902C (μmol L^−1^) + 0.0401 (R^2^ = 0.984) and a sensitivity of 89.02 μA (mmol·L^−1^)^−1^, as well as 232 μmol·L^−1^ to 2096 μmol·L^−1^ with the linear regression equation: I (μA) = 0.03720C (μmol L^−1^) + 7.724 (R^2^ = 0.978) and a sensitivity of 37.20 μA (mmol·L^−1^)^−1^, respectively.

According to the equation recommended by IUPAC, the limit of detection (LOD) was estimated to be 2.55 × 10^−8^ mol·L^−1^ and 7.01 × 10^−8^ mol L^−1^, which outperformed the sensor constructed by Pt–TiO_2_ [16], Pd/carbon black [13], CuS–RGO [38], and CuO hollow spheres [24] and equal to the nano-copper oxide obtained via electrodepositing [26] (shown in Appendix A). Based on the lower LOD with the wider linear range and the following content, we thought it was a feasible method for the determination of hydrazine hydrate in aqueous solution.

### 3.6. Anti-Interference Capability, Reproducibility, Recovery and Stability of the Sensor

To determine hydrazine in actual water samples by using the sensor, it is necessary to carry out anti-interference studies by adding hydrazine hydrate and certain substances to the 0.1 mol·L^−1^ NaOH solution in sequence. The concentration of the substances was 100 times of that of hydrazine hydrate, which was 50 μmol·L^−1^. As shown in Figure 8, the current appeared to have an obvious response to the addition of hydrazine, and there was a slight change of current when a NaNO2 solution was added; current fluctuation was almost restored to its original position after adding KCl, Na2SO4, NaAc, and NaBr solutions. Althouth the working principle of the sensor is based on oxidation of hydrazine, the adding of oxidizable component-NaNO_2_ had little effect. These results indicated that the determination of hydrazine were unimpaired by the used interferent with/without oxidizable component, which showed that the proposed sensor possesses fairly good selectivity.

For evaluating the reproducibility of the constructed sensor, five parallel modified electrodes were prepared using the same method and were investigated in the presence of 5.0 × 10^−4^ mol·L^−1^ hydrazine hydrate. The experimental results showed a relative standard deviation (RSD) of 3.23%, revealing the sensor with good reproducibility. Also, the sensor recovery was conducted as follow: after use, rinsing thoroughly under running water for 2 min, then determining whether it returns to the previous current value by CV. The current value drops slightly after using it five times. The stability in aqueous medium of the modified electrode was studied by storing it in aqueous solution at 4 °C and recording the CV daily. The change of current wasn’t obvious for a week. Two weeks later, a decrease of the current’s response (about 9.6%) for 5.0 × 10^−4^ mol·L^−1^ hydrazine was detected, which appears as good stability in aqueous medium.

### 3.7. Applications to the Water Samples

To assess the feasibility of the nano-CuO-400/GCE sensor, it was applied to analysing the water samples. The spiked recovery experiments were conducted as the standard addition method, as follow: after adding known concentrations of hydrazine into water samples (added as Ca), determining the concentration of hydrazine (added as Cd) in water samples by the constructed sensor based on the above linear equation (I = 0.08902C + 0.0401 or I = 0.0372C + 7.724), and Ca is divided by Cd for the recoveries. The amperometric response of spiked sample and the results were shown in Appendix A, respectively. The recovery rates were between 98.7% and 103.2%, and the RSDs were less than 4%, which reveals good accuracy and practicability of the proposed method. It shows the potential application to determine hydrazine hydrate in the water samples.

## 4. Conclusions

This work developed an electrochemical sensor for the rapid determination of hydrazine hydrate based on nano-CuO derived from the pyrolysis of HKUST-1 at 400 °C. The nano-CuO-400 exhibited good electrochemical properties to hydrazine hydrate in 0.1 mol·L^−1^ NaOH solution. The current increased linearly with the increase of the concentration of hydrazine within 1.98–169.3 μmol·L^−1^ and 232–2096 μmol·L^−1^ with a low detection limit of 2.55 × 10^−8^ mol·L^−1^ and 7.01 × 10^−8^ mol·L^−1^, respectively. In addition, it is possible to determine actual water samples by using this sensor. In sum, our work demonstrated that it is an option of monitoring hydrazine in aqueous phase.

## 5. Data Availability

In the article and its Appendix A, all the information including raw data were shown.

## Figures and Tables

**Figure 1 sensors-20-00140-f001:**
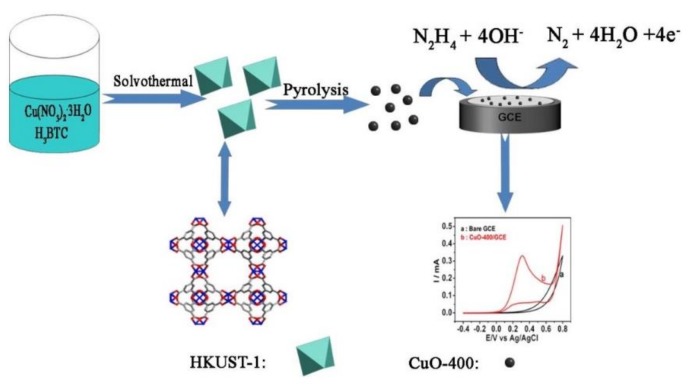
Illustration of the fabrication of nano-CuO sensor for hydrazine hydrate.

**Figure 2 sensors-20-00140-f002:**
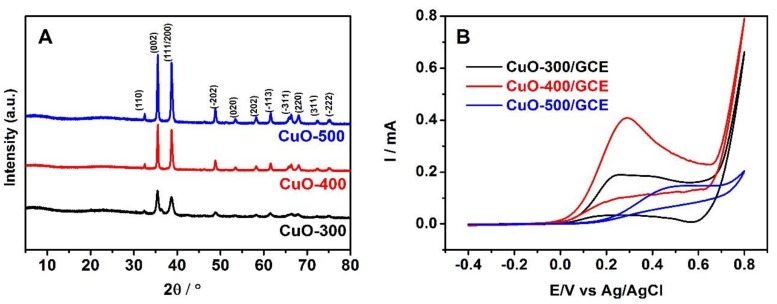
The PXRD patterns (**A**) and CV curves (**B**) of nano-CuO obtained at different temperatures.

**Figure 3 sensors-20-00140-f003:**
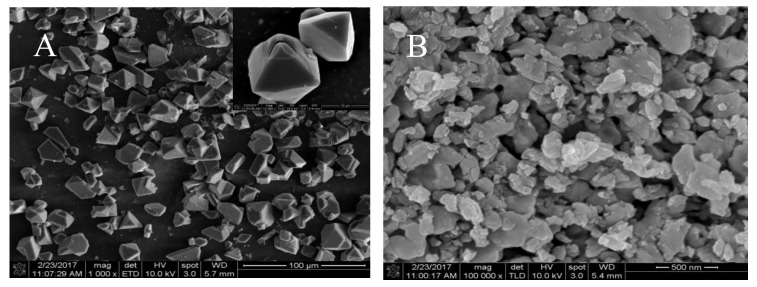
SEM images of HKUST-1(**A**) and CuO-400(**B**).

**Figure 4 sensors-20-00140-f004:**
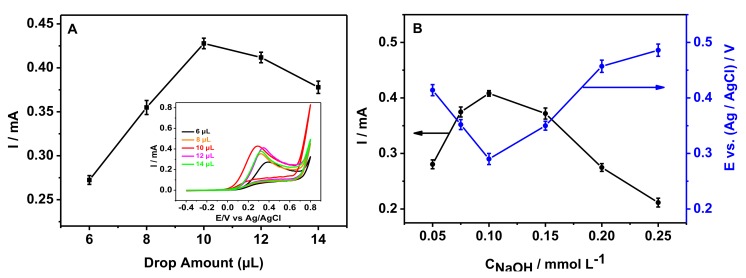
The effect of the drop amount of nano-CuO-400 (**A**)inset is the CV curves and the concentration of NaOH (**B**) on the electrochemical behavior of hydrazine.

**Figure 5 sensors-20-00140-f005:**
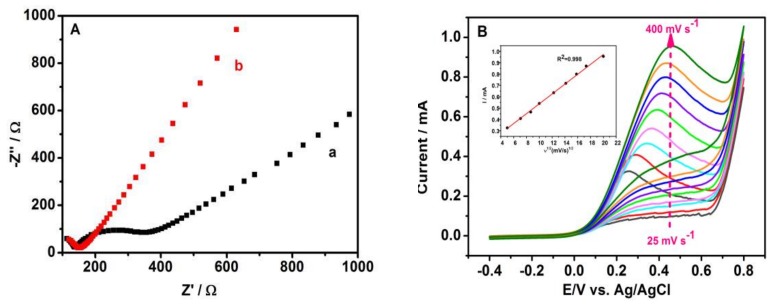
The Nyquist diagrams (**A**) of bare GCE (a) and CuO-400/GCE(b), and CV curves (**B**) of CuO-400/GCE at different scan rates (inset shows the linear relation of *I*~*v*^1/2^).

**Figure 6 sensors-20-00140-f006:**
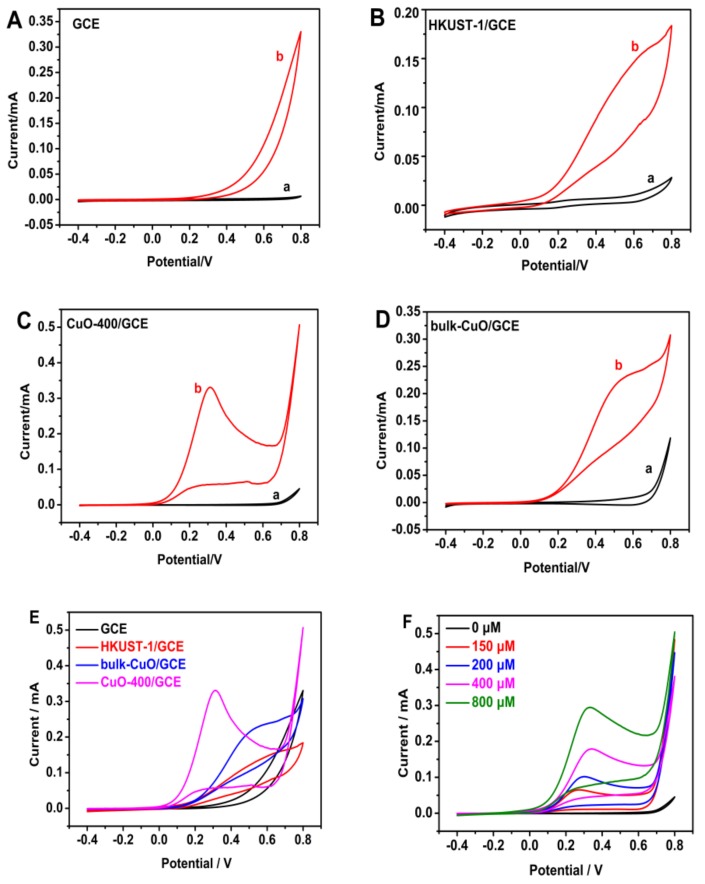
CV curves for different electrodes (**A**–**E**) and different concentrations of hydrazine hydrate at CuO-400/GCE (**F**).

**Figure 7 sensors-20-00140-f007:**
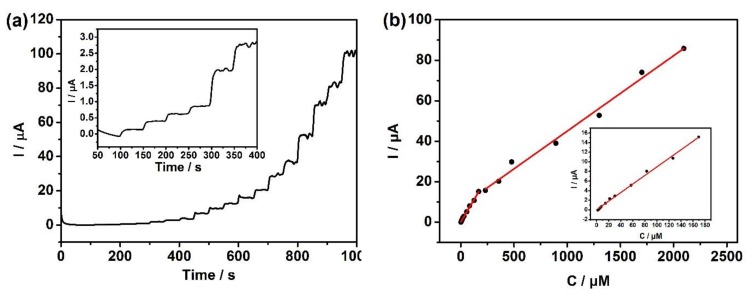
**(a**) is the amperometric *i-t* curve of the CuO-400/GCE and (**b**) shows the linear relation of *I*~C_hydrazine._

**Figure 8 sensors-20-00140-f008:**
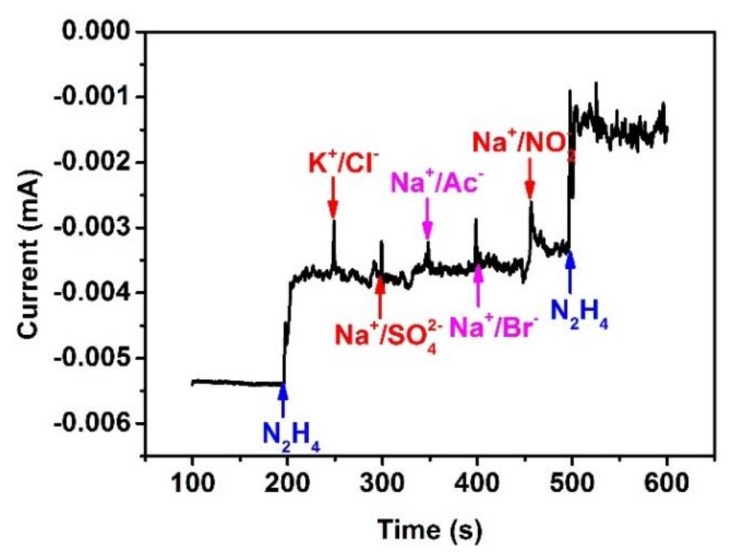
The response to the addition of hydrazine as well as interferent.

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
