# Peer review of "MOFs-Derived Nano-CuO Modified Electrode as a Sensor for Determination of Hydrazine Hydrate in Aqueous Medium"

_sensors, 2019, doi:10.3390/s20010140_

Round 1

Reviewer 1 Report

Hydrazine hydrate is highly toxic and has been listed by the US Environmental Protection Agency as a carcinogen. In this manuscript, authors constructed a sensor for the rapid determination of hydrazine hydrate based on the nano-CuO derived by controlled pyrolysis of HKUST-1 [Cu3(BTC)2(H2O)3] at 400 °C. The nano-CuO-400 exhibited excellent electrocatalytic activity toward hydrazine oxidation, with good selectivities and low detection limit. It is a good demonstration of the development of this electrochemical sensor. The manuscript is well presented. Accordingly, this work is recommended to be accepted for publication after minor revision.

There are some comments to be made and covered as follows:
1. Figure 4: Please give the deviation value of data in the figures.

2. Page 13, References, “[] U. Ragnarsson, “Synthetic methodology for alkyl substituted hydrazines,” Chemical Society Reviews, 2001, 30(4), pp 205-213.” should be “[1] U. Ragnarsson, “Synthetic methodology for alkyl substituted hydrazines,” Chemical Society Reviews, 2001, 30(4), pp 205-213”

3. Some latest data on hydrazine hydrate sensors need to be added in Table S2 in the supplementary materials for better comparison.

4. Previous reference (RSC Adv., 2013, 3, 26066-26073) reported that cellulose nanocrystals can be used as a novel support of CuO nanoparticles to improve their catalytic efficiency. The aforementioned reference can be cited in this paper.

Author Response

 a) Figure 4: Please give the deviation value of data in the figures 4.

Response:Yes, the deviation value of date was given in the figure 4.

b)Page 13, References, “[] U. Ragnarsson, “Synthetic methodology for alkyl substituted hydrazines,” Chemical Society Reviews, 2001, 30(4), pp 205-213.” should be “[1] U. Ragnarsson, “Synthetic methodology for alkyl substituted hydrazines,” Chemical Society Reviews, 2001, 30(4), pp 205-213”

Response:Yes, it was corrected.

c) Some latest data on hydrazine hydrate sensors need to be added in Table S2 in the supplementary materials for better comparison.

Response:Yes, it, Ref.S12, has been added in Table S2 (Supplementary Materials).

d) Previous reference (RSC Adv., 2013, 3, 26066-26073) reported that cellulose nanocrystals can be used as a novel support of CuO nanoparticles to improve their catalytic efficiency. The aforementioned reference can be cited in this paper.

Response: Yes, it has been cited, seen the reference [41].

Reviewer 2 Report

Yaqi Lu et al reported the electrochemical sensor for the hydrazine in aqueous solution by using MOF. Pyrolysis of HKUST-1 allowed them to get nano-CuO and coating the nano-CuO on GCE resulted the sensor (CuO/GCE). Oxidation of hydrazine was monitored by CV measurement with CuO/GCE, which resulted the LOD with 0.026 uM. In addition they demonstrated the sensor is applicable to actual water sample via spiking the hydrazine in actual water sample from Minjian river. Materials were well characterized, the sensor have great performance and manuscript is well written. Overall this reviewer believe the reader of Sensors will be interested on the manuscript. Therefore this reviewer recommend the publication of the manuscript in Sensors after minor revision.

Comment;

Working principle of the sensor is based on oxidation of hydrazine. Since many oxidizable components could perturbate the signal from the sensor. Author should describe their perspective about selectivity of the system. This reviewer believe improvement of English could increase the readability of the manuscript

Author Response

Response: Thank you for your constructive comments. The perspective about selectivity to oxidizable components has been added in Section 3.6. And we do our best to improve the readability of the manuscript.

Reviewer 3 Report

The authors report a solid study on the synthesis of nano-sized CuO and its application for hydrazine hydrate sensing in aqueous environments. Systematic experiments were performed and the majority of results is presented in a clear way. Before publication can be recommended the following points should be addressed:

Major concern: The TEM analysis is not sufficient to correlate the nanoscale morphology with sensing properties. What does a ‘good crystal structure’ (Figure 3A) imply and is there further data to support? The ‘thin layer’ mentioned in Figure 3B and 3C is not specified in terms of composition, thickness etc. and is also not apparent from the micrographs. The difference to Figure 3D is also unclear. Some more TEM data on the nanostructures would be necessary to assess the material homogeneity, e.g. size distributions or other extended analysis. The experimental data in Figure S1 and S2 should be annotated (labelling of peaks in XRD and FTIR as well as TGA data). Section 3.3: rephrase ‘different doses of nano-CuO-400 on CGE’ and ‘too thick and blocked the electronics from transmission’. Especially in the latter case a more detailed description of the sensor transduction as a function of nanomaterial thickness is needed. The authors should specify how the sensor recovery was analysed and indicate at which time scale the sensor recovers, ideally presenting some related data. The captions of figure insets are very small and should be reformatted. The authors could consider discussing potential applications of their promising nanomaterial synthesis method for applications beyond the reported electrochemical sensors by referring to recent literature on CuO nanostructures (e.g. gas sensors or Li-ion batteries).

Author Response

Response: Thank you for your constructive comments.

1) Due to the limitation of test conditions, and we just got the informations on the rough microstructure morphology of materials. The description of TEM test has been modified.

2) The experimental date in Figure S1 and S2 has been annotated in “Supplementary Materials” section.

3) The related content in Setion 3.3 has been revised.

4) The analyses methods of the sensor recovery was replenished, some related data was shown in the original Figure S3 and Table S2 

5) The captions of figure insets has been reformatted in Figure 4, Figure 5 and Figure7.

6) Description and application of CuO nanostructures has been added in “Introduction” section (page 3).

Round 2

Reviewer 3 Report

The authors have addressed some of my concerns.

However, the section related to the TEM analysis has not significantly improved, the presented data does not give much information on chemical composition, nanoscale morphology / crystal structure and material homogeneity; hence the added value to the manuscript is minimal.

The comment related to sensor recovery was related to the following: the authors studied the sensor signal for increasing concentrations. What happens if the hydrazine concentration in the water sample is decreased? Does the sensor signal also decrease again, and if yes, at what time scale? This information is important to assess if the sensor can be used multiple times. The use of the term ‘recovery’ for calculating the detected concentration can be misleading.

Author Response

Dear Prof.

With many thanks for your cordial help. We had revised it according to your suggestive comments and the responses are listed below. The corresponding revisions are labeled in red in the text.

Question 1: The section related to the TEM analysis has not significantly improved, the presented data does not give much information on chemical composition, nanoscale morphology / crystal structure and material homogeneity; hence the added value to the manuscript is minimal.

Response: Yes. The section related to the TEM has been deleted in section 2.1 and section 3.2.

Question 2: The comment related to sensor recovery was related to the following: the authors studied the sensor signal for increasing concentrations. What happens if the hydrazine concentration in the water sample is decreased? Does the sensor signal also decrease again, and if yes, at what time scale? This information is important to assess if the sensor can be used multiple times. The use of the term ‘recovery’ for calculating the detected concentration can be misleading.

Response: Sorry, the last comment related to sensor recovery is not correct. The sensor recovery in this work was conducted as follow: after use, rinsing thoroughly under running water for 2 minutes, then determining whether it returns to the previous current value by cyclic voltammetry. Generally, the sensor can be used 5 times. And the related content has been added in section 3.6.

       Thank you very much!

Sincerely yours,

Xiuling Ma et al.

Round 3

Reviewer 3 Report

No further comments.